# Strategies to prevent hospital readmission and death in patients with chronic heart failure, chronic obstructive pulmonary disease, and chronic kidney disease: A systematic review and meta-analysis

**Ryan J. Bamforth**[1,2], **Ruchi Chhibba**[1,2], **Thomas W. Ferguson**[1,2], **Jenna Sabourin**[1,2], **Domenic Pieroni**[1,2], **Nicole Askin**[1], **Navdeep Tangri**[1,2], **Paul Komenda**[1,2], **Claudio Rigatto**[1,2]*

1 Max Rady College of Medicine, University of Manitoba, Winnipeg, Manitoba, Canada, 2 Seven Oaks Hospital Chronic Disease Innovation Centre, Winnipeg, Manitoba, Canada

* crigatto@sbgh.mb.ca

## Abstract

### Background

Readmission following hospital discharge is common and is a major financial burden on healthcare systems.

### Objectives

Our objectives were to 1) identify studies describing post-discharge interventions and their efficacy with respect to reducing risk of mortality and rate of hospital readmission; and 2) identify intervention characteristics associated with efficacy.

### Methods

A systematic review of the literature was performed. We searched MEDLINE, PubMed, Cochrane, EMBASE and CINAHL. Our selection criteria included randomized controlled trials comparing post-discharge interventions with usual care on rates of hospital readmission and mortality in high-risk chronic disease patient populations. We used random effects meta-analyses to estimate pooled risk ratios for all-cause and cause-specific mortality as well as all-cause and cause-specific hospitalization.

### Results

We included 31 randomized controlled trials encompassing 9654 patients (24 studies in CHF, 4 in COPD, 1 in both CHF and COPD, 1 in CKD and 1 in an undifferentiated population). Meta-analysis showed post-discharge interventions reduced cause-specific (RR = 0.71, 95% CI = 0.63–0.80) and all cause (RR = 0.90, 95% CI = 0.81–0.99) hospitalization, all-cause (RR = 0.73, 95% CI = 0.65–0.83) and cause-specific mortality (RR = 0.68, 95% CI

**Data Availability Statement:** All relevant data are within the paper and its Supporting Information files.

**Funding:** We gratefully acknowledge the support of the University of Manitoba Max Rady College of Medicine.

**Competing interests:** The authors have declared that no competing interests exist.

= 0.54–0.84) in CHF studies, and all-cause hospitalization (RR = 0.52, 95% CI = 0.32–0.83) in COPD studies. The inclusion of a cardiac nurse in the multidisciplinary team was associated with greater efficacy in reducing all-cause mortality among patients discharged after heart failure admission (HR = 0.64, 95% CI = 0.54–0.75 vs. HR = 0.87, 95% CI = 0.73–1.03).

## Conclusions

Post-discharge interventions reduced all-cause mortality, cause-specific mortality, and cause-specific hospitalization in CHF patients and all-cause hospitalization in COPD patients. The presence of a cardiac nurse was associated with greater efficacy in included studies. Additional research is needed on the impact of post-discharge intervention strategies in COPD and CKD patients.

## Introduction

Readmission following hospital discharge is a common occurrence and results in a major financial burden on health care systems [1]. In particular, patients suffering from chronic diseases, mainly chronic obstructive pulmonary disease (COPD) and congestive heart failure (CHF), have frequent readmissions and suffer from increased morbidity and mortality related to these events [2]. Overall, up to 19.6% of patients admitted with chronic health conditions in the United States are readmitted within one month and 34.0% within three months of discharge [3].

It has been estimated that nearly 60% of hospital readmissions are preventable [4]. Risk factors associated with avoidable readmissions include those related to patient, social, clinical and system factors such as patient behaviors, community services, adequacy and appropriateness of assessment and treatment, as well as accessibility and coordination within the healthcare delivery system [4]. Preventing such hospitalizations may result in up to $12 billion dollars of savings in the United States alone [5]. The absence of suitable post-discharge care is just one of many potential factors contributing to future hospitalizations [6] and continuity of care in high-risk chronic disease patients is crucial in order to mitigate the risk of readmission [7].

A variety of post-discharge interventions have been proposed to reduce this readmission risk, ranging from minimal (i.e. follow-up telephone calls), to complex, multifaceted interventions such as "virtual wards", which provide patients with a period of intensive multidisciplinary team management, often employing telemonitoring and nurse led case-management strategies [8]. Use of virtual wards following hospital discharge has been associated with a lower risk of readmission in certain health conditions. In a recent systematic review, we found that virtual wards were effective in improving clinical outcomes and reducing hospital readmissions in heart failure [9]. However, the costs and complexities associated with virtual wards may be a barrier to implementation. The question of whether, and to what degree, less complex and less costly interventions can be as effective as highly complex virtual wards in minimizing hospital readmissions and death in high-risk chronic disease patient populations is unknown.

To address this knowledge gap, we conducted a systematic review of randomized clinical trials examining different follow-up programs and strategies specific to high-risk chronic disease populations after hospital discharge. These interventions ranged from simple (e.g.,

telephone calls) to more complex and costly programs such as virtual wards. We focused primarily on chronic kidney disease (CKD), heart failure (HF), and chronic obstructive pulmonary disease (COPD). We chose these conditions because all of them are common and associated with frequent and costly exacerbations/acute decompensations necessitating admission for stabilization and associated with high risk of recurrence [2, 3, 10–12]. Our co-primary objectives were: 1) to estimate the efficacy of post-discharge interventions in each chronic disease reviewed, and 2) identify intervention components most strongly associated with efficacy.

## Materials and methods

This systematic review is reported in accordance with the Preferred Reporting Items for Systematic Reviews and Meta-Analysis (PRISMA) Statement [13]. It was performed under a pre-written protocol that was not registered with a traditional systematic review registry.

### Data sources and searches

We searched MEDLINE, PubMed, Cochrane, EMBASE and CINAHL identifying relevant studies published up to November 2020. The goal of our literature search was to identify all relevant studies in CHF, COPD and CKD that included any post-discharge intervention. In collaboration with a medical librarian, we developed search terms tailored to each database. We incorporated keywords such as "virtual ward", "telemedicine," "case management", and other types of interventions to maximize the capture of potentially relevant studies from the available literature. The complete search strategy can be found in supplemental materials (S1 Table).

### Study selection

We included randomized controlled trials comparing post-discharge interventions with usual care in community-dwelling adult patients (≥18 years of age) recruited to the trial directly following or within three months of hospitalization. The main outcomes of interest were rates of hospital readmission and mortality. No restrictions were placed on dates or language. Two independent reviewers (JS, DP) screened the titles and abstracts of all articles identified in the database search. Potentially eligible articles underwent independent full text review by the same two reviewers to identify the final set of articles. Included articles went on to data extraction. Reasons for exclusion were documented for the remaining articles. Disagreements about inclusion were settled by consensus, with the assistance of a third reviewer (CR, RC) as necessary.

### Data extraction

Relevant data, including first author, year of publication, study location (country), sample size, study population, mean age, proportion of women, study design including components associated with risk of bias, description of usual care, as well as description of intervention(s) employed were extracted from the included studies. All-cause hospitalization, cause-specific hospitalization, all-cause mortality, and cause-specific mortality for both control and intervention groups were also extracted from studies. All data was abstracted in duplicate (JS, DP). Conflicts were resolved by consensus, or by a third reviewer if consensus was not reached (CR, RC).

### Classification of intervention strategies

Based on our review, we identified five types of post-discharge intervention strategies that were variably incorporated into a given intervention. These strategies included pre-discharge

disease specific patient education, post-discharge follow-up telephone calls, home visits, continuous or semi-continuous telemonitoring of vital signs, and coordinated multidisciplinary team care. For the purposes of our review, multidisciplinary team care was defined as care offered by at least three health care providers in three different areas of patient care. We also captured information on whether a given team member was described as having (or not having) any disease specific training or expertise (e.g., heart failure specialist vs. general internist; Heart failure nurse vs. general nurse). Patient education was deemed to be additional instruction or training specific to self-management of the disease in question, beyond what is generally provided to patients upon discharge. Telemonitoring was defined as continuous or semi-continuous monitoring of patient data via a dedicated or web-based platform. There was no inclusion threshold placed on number telephone calls, home visits, or telemonitoring frequency. Additionally, telephone calls, home visits, and telemonitoring could be conducted by any care provider. In most studies, these interventions were overseen or applied by nurses. For our analysis, we recorded the number and type(s) of intervention(s) used in each study.

## Quality assessment

Risk of bias was determined via the application of the Cochrane Collaboration's Tool (CCT) for assessing risk of bias in randomized trials [14, 15]. This tool considers 6 domains of bias (selection bias, performance bias, detection bias, attrition bias, reporting bias, and other bias). Studies were classified as being at low, unclear, or high risk of bias in each of the 6 bias domains. The evaluation was conducted by two reviewers (JS, DP), with discrepancies resolved by consensus (CR, RC) as described for study inclusion above. Further information on the risk of bias criteria is located in supplemental materials (S2 Table).

## Data synthesis and analysis

We used random effects meta-analyses to estimate pooled risk ratios and 95% confidence intervals for all-cause and cause-specific mortality as well as all-cause and cause-specific hospitalization for each chronic disease category [16]. To address our pre-hoc hypothesis that number, type, or specificity of intervention strategies used might influence effectiveness, we conducted subgroup analyses to: 1) compare efficacy in studies using fewer or greater than the median number of strategies; 2) compare studies according to type of intervention strategy used; 3) compare studies with disease specific teams. These latter analyses were only performed for the heart failure studies, as the number of COPD and CKD studies was too small. In a post hoc, secondary analysis, we created a simple scoring system to rank intervention strategies in terms of complexity: pre-discharge patient education = 1; post-discharge telephone calls = 2; home visits = 3; continuous tele-monitoring = 4; and multidisciplinary team care = 5. We defined a complexity score for a study as the simple sum of the complexity ranks of all strategies employed in that study. We then used meta-regression to examine whether complexity as measured using this score was associated with efficacy of the intervention. Additional sensitivity analyses were performed by excluding studies at high risk of reporting, attrition, selection or concealment bias to assess uncertainty. Finally, to examine whether there is evidence of an era effect, we conducted a subgroup analysis including only recently published heart-failure studies (since January 2009).

# Results

## Study selection

A flow diagram summarizing our literature search and selection of relevant articles is shown below in Fig 1. A total of 8690 articles were identified based on our search strategy.

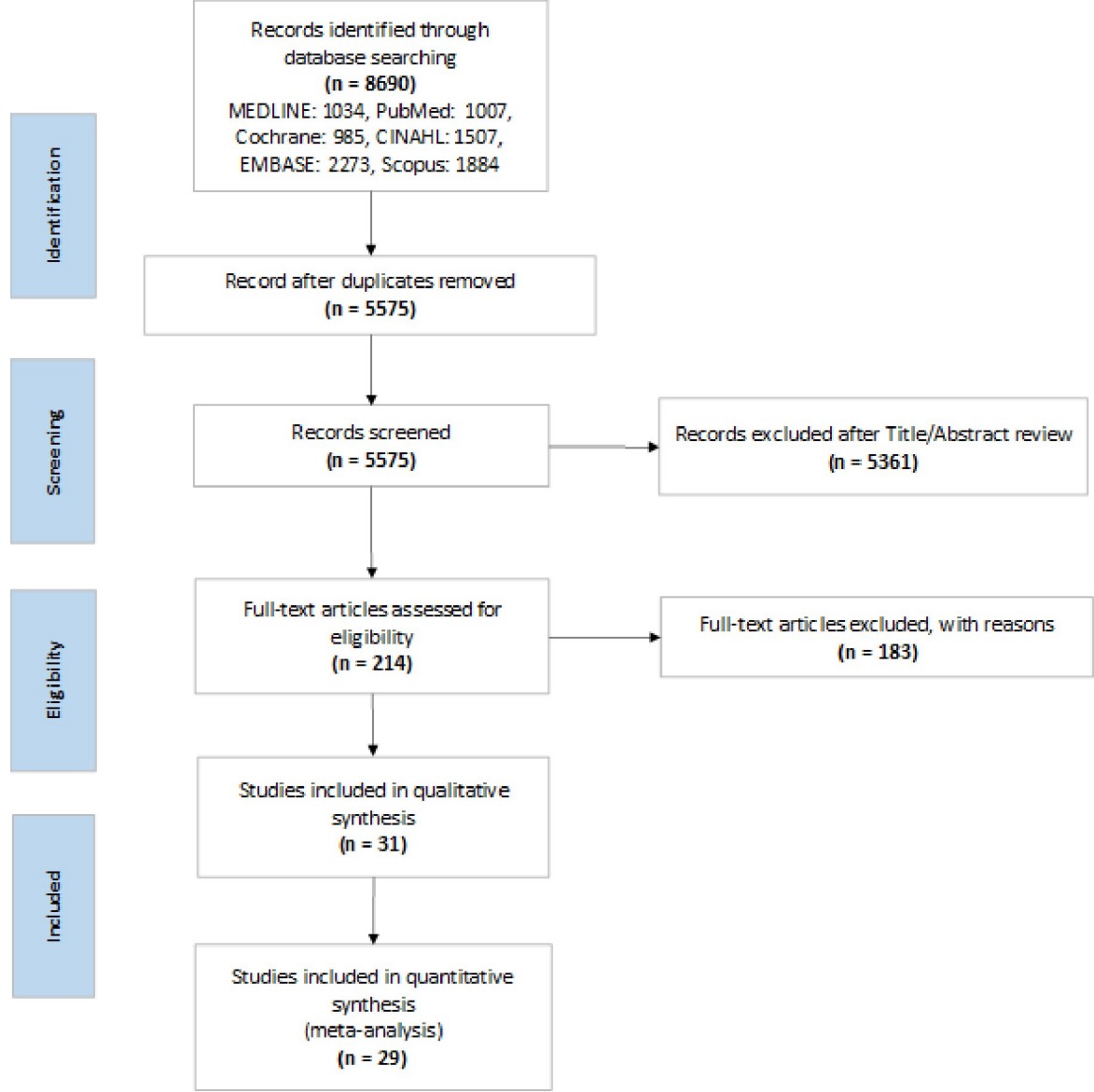

**Fig 1. PRISMA flow diagram of the study selection process for the systematic review.**

The titles and abstracts of these articles were screened, and 214 were selected for full text review. Of these 214 articles, 31 met all the inclusion criteria. A summary of the full-text review is located in supplemental materials (S3 Table).

## Characteristics of selected studies

We included 31 randomized controlled trials encompassing 9654 patients in our systematic review [17–47]. The mean age of the various study populations ranged from 56 to 80 years. The study populations consisted of patients admitted to hospital within the past three months due to HF, CKD, or COPD, and discharged home. Most studies had a larger proportion of males than females. Of the included studies, 24 were specifically on congestive heart failure

[17–37, 44, 46, 47], and 1 study included both congestive heart failure and chronic obstructive pulmonary disease [38]. There were 4 studies specifically on chronic obstructive pulmonary disease [39–41, 45], 1 on chronic kidney disease [42], and 1 study which focused on undifferentiated, high risk chronic disease patient populations [43]. The last study could not be included because separate outcome data for each chronic disease of interest was not reported [43].

In all studies, the control group consisted of patients receiving "usual care" following hospital discharge. The definition of usual care varied across studies, however. Typically, usual care was comprised of an outpatient follow-up appointment scheduled with the patient's primary care physician. However, some studies described more intensive usual care practices, such as follow-up with a disease focused team.

The interventions described also varied between studies. However, all included one or more of the following elements: patient education, telephone calls, home visits, telemonitoring, and multidisciplinary care. The intervention period varied greatly between studies, as did the follow-up period (from six months to two years). Additional details on all included studies can be found in Tables 1 and 2.

## Quality of reporting and risk of bias

Risk of bias for the included studies is summarized in the supplemental material (S4 Table). Most studies were judged to be at low risk of reporting, detection, and selection biases. Blinding of participants was not attempted in any of the studies. While this is understandable given the nature of the intervention, lack of participant blinding does increase the risk of performance bias in all studies. Moreover, a description of steps taken to ensure blinding of outcome assessors was not included in many studies, leading to a high overall risk of measurement bias in the included studies. Overall, three of the thirty-one studies were deemed at high risk of bias [26, 33, 38]. In sensitivity analyses, the calculated risk ratios were found to be of similar direction and magnitude when studies at high risk of bias were excluded (S1–S3 Figs).

## Qualitative analysis

Two of the thirty-one included studies were excluded from the meta-analysis. The first study was excluded as it was the sole study in CKD patients that met inclusion criteria and we were unable to meta-analyze it on its own [42] The second study was excluded due to the inability to extract outcome data for each separate chronic disease of interest [42, 43]. Li et al. examined the effectiveness of telephone support in patients undergoing peritoneal dialysis in China [42]. Overall, 135 patients were recruited (66 randomized to control group, 69 to intervention group) with results showing a statistically significant difference in quality of life measures (symptom/problem, work status, staff encouragement, patient satisfaction and energy/fatigue as measured by the KDQOL-SF) favoring the intervention group, with no differences in blood chemistry and complication control as well as health-care utilization (readmissions) between the control and intervention groups [42]. Dhalla et al. evaluated the effectiveness of virtual wards on readmissions and death in high-risk patients (based upon length of stay, acuity of the admission, comorbidities, and emergency department visits in the previous 6 months) [43]. Patients randomized to the intervention group were admitted to the virtual ward once discharged, providing them telephone access to a multidisciplinary team consisting of a case coordinator, part-time pharmacist, part-time nurse, or nurse-practitioner, full-time physician, and a clerical assistant, in addition to usual care. The authors found that virtual wards had no statistically significant effect on readmissions and death at 30 and 60 days, 6 months and 1 year post discharge in comparison to usual care [43].

**Table 1. Overview of study characteristics.**

| First author, YR | Country | N | Period of Intervention (months) | Follow-Up Period (months) | Description of Usual Care | Description of Intervention | Patient Population | Mean Age of Total Study Population | % Female in Total Study Population |
|---|---|---|---|---|---|---|---|---|---|
| *Heart Failure Patient Population* | | | | | | | | | |
| Stewart, 1999 [37] | Australia | 97 | 0.25 (1 week) | 18 | Patients received usual post-discharge care. | Patients received one home visit by a nurse and a pharmacist shortly after discharge and were referred to their primary care provider, as necessary. | Patients hospitalized with heart failure and with a history of at least 1 addition hospitalization for acute heart failure. | 75 | 52 |
| Stewart, 1999 [36] | Australia | 200 | 6 | 6 | Usual care was comprised of both community-based and inpatient contact with a cardiac rehabilitation nurse, dietitian, social worker, pharmacist, and community nurse. Patients had follow-up appointments with their primary care physician and/ or a cardiologist within 2 weeks of being discharged. | In addition to usual care, patients received a home visit by a cardiac nurse 1–2 weeks following discharge. Based on the nurse's assessment and after consultation with the patient's primary care physician and cardiologist, modifications to the treatment plan took place if necessary. Patients received a second home visit if they had 2 or more hospital readmissions within 6 months. All patients were contacted by telephone at 3 months and 6 months. | Patients $\geq$ 55 years of age admitted with heart failure (LVEF $\leq$ 55%, NYHA Class 11-1V), and at least 1 previous hospitalization for acute heart failure. | 75.7 | 38 |
| Kasper, 2002 [35] | USA | 200 | 6 | 6 | Patients received follow-up care from their primary care physician. Each primary care physician received a baseline management plan from the heart failure cardiologist as documented in their patient's chart. | Patients received follow-up care from a multidisciplinary team consisting of a nurse coordinator, heart failure nurse, cardiologist, and the patient's primary care physician. Patients were contacted via telephone by the nurse coordinator within 3 days of discharge, then once per week for one month, twice in the second month, and then once per month until the end of the intervention period. Additionally, patients were scheduled for appointments with a heart failure nurse at a clinic or in their home (rare). The multidisciplinary team met weekly to discuss patient care and optimize treatment. Patients were also given a contact number that they could call 24 h/day. | Patients hospitalized with NYHA Class III/IV heart failure and the having at least one additional designated high-risk criteria: age >70 years, LVEF <35%, one or more additional heart failure admission in the past year, ischemic cardiomyopathy, peripheral edema upon hospital discharge, <3 kg weight loss during hospital stay, peripheral vascular disease or hemodynamic findings of pulmonary capillary wedge pressure >25 mm Hg, cardiac index <2.0 l/min/m$^2$, systolic blood pressure >180 mm Hg or diastolic blood pressure >100 mm Hg. | 63.5 | 39.5 |
| Goldberg, 2003 [34] | USA | 280 | 6 | 6 | Prior to discharge, all patients received education regarding heart failure. Patients in the usual care group were then followed-up by their primary care provider. | In addition to usual care, patients received follow-up care via telemonitoring equipment including an electronic weight scale and symptom response system. Everyday, patients weighed themselves and answered questions related to heart failure symptoms. This information was automatically transmitted to the research nurses for interpretation. Increases in weight or concerning symptoms were reported to the patient's physician. | Patients hospitalized with NYHA class III or IV heart failure and LVEF $\leq$ 35%. | 59 | 32 |

*(Continued)*

**Table 1.** (Continued)

| First author, YR | Country | N | Period of Intervention (months) | Follow-Up Period (months) | Description of Usual Care | Description of Intervention | Patient Population | Mean Age of Total Study Population | % Female in Total Study Population |
|---|---|---|---|---|---|---|---|---|---|
| **Dunagan, 2005** [33] | USA | 151 | 12 | 12 | Patients received usual post-discharge care. | Patients received a telephone call from a study nurse within 3 days of hospital discharge and then once per week for 2 weeks. Frequency of telephone calls were subsequently individualized based on patient needs. The goal of these telephone calls was to improve self-management, lifestyle, and adherence to treatment plan. | Patients $\geq$21 years of age hospitalized with NYHA class II-IV heart failure. | 70 | 56 |
| **Cleland, 2005** [32] | Germany, Netherlands, UK | 253 | 15 | 15 | Patients' management plans were sent to their primary care provider and patients were followed-up there. | In addition to usual care, patients were followed-up via telemonitoring equipment including an electronic weight scale, an automated sphygmomanometer, and an ECG. Patients took their measurements twice per day and they were automatically sent to a web server at each research site. Measurements that fell outside a predetermined range were analyzed by the study nurse and corrective action was taken, as necessary. | Patients hospitalized >48 hours due to worsening heart failure within the past 6 weeks, with persistent symptoms, LVEF <40%, an LV end- diastolic dimension >30 mm/m (height). Patients also had to be receiving furosemide at a dose of $\geq$40 mg/day or equivalent (e.g., $\geq$1 mg of bumetanide or $\geq$10 mg of torasemide). Additionally, patients had to have at least one of these high-risk criteria: unplanned cardiovascular admission >48 hours within the past 2 years, LVEF <25%, or treatment with furosemide at a dose of $\geq$100 mg/day or equivalent. | 67.5 | 19 |
| **Cleland, 2005** [32] | Germany, Netherlands, UK | 258 | 15 | 15 | Patients' management plans were sent to their primary care provider and patients were followed-up there. | In addition to usual care, patients were contacted by a heart failure nurse once per month. Patients could also call the nurse at anytime to address questions and/or concerns. | Patients hospitalized >48 hours due to worsening heart failure within the past 6 weeks, with persistent symptoms, LVEF <40%, an LV end- diastolic dimension >30 mm/m (height). Patients also had to be receiving furosemide at a dose of $\geq$40 mg/day or equivalent (e.g., $\geq$1 mg of bumetanide or $\geq$10 mg of torasemide). Additionally, patients had to have at least one of these high-risk criteria: unplanned cardiovascular admission >48 hours within the past 2 years, LVEF <25%, or treatment with furosemide at a dose of $\geq$100 mg/day or equivalent. | 67.5 | 23 |

(Continued)

**Table 1.** (Continued)

| First author, YR | Country | N | Period of Intervention (months) | Follow-Up Period (months) | Description of Usual Care | Description of Intervention | Patient Population | Mean Age of Total Study Population | % Female in Total Study Population |
|---|---|---|---|---|---|---|---|---|---|
| **Del Sindaco, 2007** [31] | Italy | 173 | 24 | 24 | Patients received discharge summaries and educational information regarding lifestyle modifications related to heart failure and self-monitoring. Following discharge, patients received usual care from their primary care provider and/or cardiologist. Additionally, vital signs and events were reported by telephone calls every 6 months. | The patient follow-up team consisted of a cardiologist specialized in geriatrics, 2–4 specialized nurses, and the patient's primary care physician. Patients were provided with written recommendations, a weight chart, a telephone number available 6 hours per day, and information brochures. Post-discharge follow-ups occurred at the heart failure clinics within 1–2 weeks of discharge and again at 1, 3, and 6 months afterwards. Additionally, nurses made follow-up telephone calls and were directly involved in educating patients. | Patients ≥70 years of age being discharged following hospitalization due to heart failure (NYHA Class 11-IV) lasting ≥24 hours. | 77 | 48 |
| **Wakefield, 2008** [29] | USA | 101 | 3 | 6 | Patients were followed-up with their primary care provider. | Patients were contacted via video calls 3 times in the week following discharge, and then weekly for 11 weeks. Patients had a symptoms checklist, weight scale, blood pressure cuff, and tape measure to monitor fluid overload. Patients were assessed throughout the study by the study nurse and, together with the patient's physician, the study nurse made adjustments as needed. Throughout the intervention, nurses also provided patients with behavioural skill strategies to improve self-efficacy, self-monitoring, and goal setting. | Patients admitted to hospital for heart failure. | 68.1 | 2 |
| **Wakefield, 2008** [29] | USA | 96 | 3 | 6 | Patients were followed-up with their primary care provider. | Patients were contacted via telephone 3 times in the week following discharge, and then weekly for 11 weeks. Patients had a symptoms checklist, weight scale, blood pressure cuff, and tape measure to monitor fluid overload. Patients were assessed throughout the study by the study nurse and, together with the patient's physician, the study nurse made adjustments as needed. Throughout the intervention, nurses also provided patients with behavioural skill strategies to improve self-efficacy, self-monitoring, and goal setting. | Patients admitted to hospital for heart failure. | 69.5 | 1 |
| **Antonicelli, 2008** [30] | Italy | 57 | 12 | 12 | Prior to discharge, patients and/or their family members/caregivers attended educational sessions regarding heart failure and lifestyle. Subsequently, patients received follow-up care in the form of scheduled appointments at the heart failure outpatient clinic at least every 4 months and more as needed. | In addition to usual care and following discharge, the heart failure team contacted patients at least weekly to gather information on symptoms, adherence to treatment, blood pressure, heart rate, body weight, and 24-hour urine output. Patients were also required to have an ECG once per week. Based on this information, treatment was adjusted as needed. Patients also regularly attended the heart failure clinic. | Patients ≥70 years of age admitted to hospital with worsening signs and symptoms of heart failure. | 78 | 39 |

(*Continued*)

**Table 1.** (Continued)

| First author, YR | Country | N | Period of Intervention (months) | Follow-Up Period (months) | Description of Usual Care | Description of Intervention | Patient Population | Mean Age of Total Study Population | % Female in Total Study Population |
|---|---|---|---|---|---|---|---|---|---|
| **Giordano, 2009** [28] | Italy | 460 | 12 | 12 | All patients received pre-discharge education and then were referred to their primary care provider for a follow-up appointment within 2 weeks of discharge. | In addition to usual care, patients were given a portable device for telemonitoring which included an ECG. Scheduled telemonitoring was done every week or every 15 days. This consisted of the nurse assessing the patient by asking questions regarding symptoms, lifestyle, etc. Patients could also call the study nurse with any questions and/or concerns. Once per week, nurses and cardiologist met to discuss patients enrolled in the study and their care. | Patients hospitalized with heart failure, LVEF <40%, and at least 1 hospitalization due to heart failure within the past year. | 57 | 15 |
| **Scherr, 2009** [26] | Austria | 120 | 6 | 6 | Patients received regular follow-up care with their primary care physician. | Patients were provided with telemonitoring equipment including a mobile phone, a weight scale, and a sphygmomanometer for an automated measurement of blood pressure and heart rate. Everyday, patients took their vital signs and entered them into the mobile phone's Internet browser where they could be interpreted by study physicians. Physicians contacted patients should their treatment plan need to be adjusted based on their measurements. | Patients with decompensated heart failure admitted to hospital for >24 hours within the past month. | 66 | 27 |
| **Dar, 2009** [27] | England | 182 | 6 | 6 | Initially, patients received a home visit by the study nurse during which time they were given advice regarding self-monitoring of heart failure. Afterwards, they were provided follow-up care from a secondary care-based heart failure service, including at least one cardiologist or physician and a heart failure nurse, via regular clinic appointments with the heart failure team. The frequency of follow-up appointments was decided by the heart failure team. Telephone support was also available to patients during regular business hours. | In addition to usual care, patients received home telemonitoring via an electronic weight scale, automated blood pressure cuff, pulse oximeter, and a control box connected to their home phone line. Every morning, patients were given instructions to use this equipment to measure their vital signs and answer four questions regarding symptoms of heart failure decompensation. This information was automatically transmitted to the hospital base station where it was reviewed by a heart failure nurse. Changes in vital signs from pre-set parameters triggered an alert and patients were contacted, resulting in one of the following: further lifestyle or medication advice, recommendation to contact primary care, or early review in secondary care. | Patients >18 years of age hospitalized with heart failure and in NYHA Class II-IV upon discharge. | 72 | 34 |
| **Leventhal, 2011** [25] | Switzerland | 42 | 12 | 12 | The usual group were followed-up by their primary care provider. | Patients received a home visit from a heart failure nurse ~1 week after discharge and then 17 telephone calls throughout the intervention to address patient questions and/or concerns, which the nurse discussed with the patient's primary care provider. The home visit consisted of an assessment, patient education, and goal setting. | Patients hospitalized with decompensated heart failure (NYHA class II-IV). | 77 | 38.1 |

(*Continued*)

**Table 1.** (Continued)

| First author, YR | Country | N | Period of Intervention (months) | Follow-Up Period (months) | Description of Usual Care | Description of Intervention | Patient Population | Mean Age of Total Study Population | % Female in Total Study Population |
|---|---|---|---|---|---|---|---|---|---|
| **Dendale, 2012** [23] | Belgium | 160 | 6 | 6 | Prior to discharge, all patients and their family members participated in an educational session with a heart failure nurse. Patients were seen at the heart failure clinic 2 weeks post-discharge. Thereafter, patients were followed-up by their primary care provider. | Following discharge, patients received telemonitoring via an electronic body weight scale and a blood pressure monitoring device which were connected to a cell phone via Bluetooth. Patients used this equipment daily and the information was automatically transmitted to a central computer. The heart failure clinic and the patient's primary care physician were alerted if the measurements fell outside of a predetermined range. The primary care physician then contacted the patient to make the necessary adjustments. The heart failure nurse also phoned the patient 1–3 days after the alert. Additionally, patients were scheduled for follow-up in the heart failure clinic 2 weeks, 3 months, and 6 months following discharge, but could visit the clinic more often. | Patients hospitalized for fluid overload due to heart failure and required to start or increase diuretic treatment. | 76 | 35 |
| **Angermann, 2012** [24] | Germany | 715 | 6 | 6 | Patients received treatment plans, discharge letters, and follow-up appointments with either a primary care provider or cardiologist within 1–2 weeks following discharge. | In addition to usual care, patients and relatives participated in pre-discharge educational sessions with a specialist nurse. Following discharge, patients received telephone-based monitoring involving symptom and well-being questionnaires, planning of medications in conjunction with primary care providers, individualized specialist care coordinated by nurses with the patients' physician, and further education. During the first month following discharge, patients received telephone calls once per month. The frequency of phone calls were subsequently individualized based on patient needs. | Patients ≥18 years of age hospitalized with signs and symptoms of decompensated heart failure and LVEF ≤ 40%. | 68.6 | 29 |
| **Tsuchihashi-Makaya, 2013** [22] | Japan | 161 | 6 | 12 | Prior to discharge, patients received extensive education by a cardiologist, nurse, dietitian, and pharmacist. Patients then had regular post-discharge follow-up care including management by the cardiologist. | In addition to usual care, follow-up care included home visits within 2 weeks of discharge and then once every 2 weeks for the following month and a half. Thereafter, nurses made monthly telephone calls to patients until 6 months following discharge. During all encounters, nurses assessed potential symptoms of decompensated heart failure, body weight, and educated patients. Nurses consulted a multidisciplinary team consisting of a cardiologist, dietitian, pharmacist, and social worker throughout the study. | Patients hospitalized with heart failure. | 76 | 30 |

(Continued)

**Table 1.** (Continued)

| First author, YR | Country | N | Period of Intervention (months) | Follow-Up Period (months) | Description of Usual Care | Description of Intervention | Patient Population | Mean Age of Total Study Population | % Female in Total Study Population |
|---|---|---|---|---|---|---|---|---|---|
| **Villani, 2014** [21] | Italy | 80 | 12 | 12 | Prior to discharge, all patients participated in an educational session with a heart failure nurse. Following discharge, patients were scheduled for appointments at a heart failure clinic every three months. | In addition to usual care, patients were given a handheld PDA to measure their heart rate, body weight, blood pressure, and ECG. Everyday, patients were alerted and reminded to measure and send this information in at predetermined times and frequencies. The PDA software also included a questionnaire regarding mental wellbeing and medications as well as reminders to take their medications at the correct time. A cardiologist interpreted the information and contacted patients with modifications or recommendations as needed. A psychologist was also available to patients during their follow-up visits in clinic. | Patients hospitalized with NYHA class III/IV heart failure, LVEF <40%, and at high risk of readmission defined by 2 or more of the following: age >70, >2 hospitalizations related to heart failure in the past 6 months, >1 comorbidities (diabetes, COPD, cerebrovascular disease, renal failure). | 72 | 27.5 |
| **Yu, 2015** [20] | Hong Kong | 178 | 9 | 9 | Patients were scheduled for follow-up at a specialist clinic 4–6 weeks following discharge. | Patients received pre-discharge visits as well as two post-discharge home visits from a cardiac nurse which involved individualized education and goal setting regarding self-care. Patients received telephone calls from the cardiac nurse 1 week following the second home visit, then every 2 weeks for three months, followed by every 2 months for 6 months. Additional home visits were conducted for patients requiring further assessment and care. Patients were also provided with a telephone number that they could call during regular business hours to have questions and/or concerns addressed by the cardiac nurse. | Patients ≥60 years of age admitted to hospital due to heart failure. | 78.6 | 55 |
| **Ritchie, 2016** [38] | USA | 346 | 3 | 3 | Patients received usual post-discharge care by their primary care provider. | Prior to discharge and during the follow-up period, patients received support and education from care transition nurses and an "E-Coach" (IVR-supported care system). This device provided automated telephone calls and was used to accumulate date on patients, provide patients with individualized management plans, and alert the nurse when intervention was required. Patients received calls from the E-Coach everyday for a week following discharge and either everyday or every 3 days for an additional 21 calls. | Patients admitted to hospital with either heart failure being discharged home. | 63.3 | 48.7 |

(*Continued*)

Table 1. (Continued)

| First author, YR | Country | N | Period of Intervention (months) | Follow-Up Period (months) | Description of Usual Care | Description of Intervention | Patient Population | Mean Age of Total Study Population | % Female in Total Study Population |
|---|---|---|---|---|---|---|---|---|---|
| **Ong, 2016** [19] | USA | 1437 | 6 | 6 | Patients received extensive pre-discharge education and often a follow-up telephone call once discharged home. They then received routine follow-up care. | Patients received pre-discharge education regarding heart failure, 9 telephone calls over the period of 6 months following discharge (the first one being 2–3 days after discharge, then once per week for the first month, followed by once per month for the remainder of the study), and home telemonitoring via a wireless transmission pod, a weight scale, and a blood pressure and heart rate monitor. Patients also used these equipment to answer 3 questions regarding heart failure symptoms. This information was automatically transmitted back to central servers via Bluetooth for interpretation by study nurses. Concerning symptoms or measurements that fell outside a predetermined range triggered an alarm and patients were contacted. | Patients ≥50 years of age hospitalized for decompensated heart failure. | 73 | 46.2 |
| **Kotooka, 2018** [17] | Japan | 181 | 12 | 12 | Upon discharge, patients were provided education and recommended to measure their body weight daily. | Patients were monitored via an electronic scale with a body composition meter, a sphygmomanometer, and a receiver which automatically transmitted blood pressure, heart rate, body weight, and body composition data to the central server. Nurses monitored the incoming data and if it exceeded a pre-set threshold, the patient's primary care provider was notified, and corrective action was taken. | Patients ≥20 years of age hospitalized due to NYHA class II-IV decompensated heart failure and being discharged home. | 66.3 | 40.9 |
| **Olivari, 2018** [18] | Italy | 339 | 12 | 12 | Patients received routine post-discharge care, which was generally comprised of a follow-up visit within the first month of discharge. | Patients were given wearable wrist devices which they used to measure their heart rate, blood pressure, ECG, and pulse oximetry, as well as a digital weight scale used to measure their weight. This information was transmitted every day, 5 days per week for interpretation and physicians were notified of abnormal or concerning results. | Patients ≥65 years of age hospitalized with heart failure in the past 3 months and LVEF <40% or >40% with BNP >400 (or NT-proBNP >1500) during hospitalization. | 80.3 | 36.8 |
| **Wierzchowiecki, 2006** [44] | Poland | 160 | 12 | 12 | Regular follow-up with their primary care physician. | Patients followed-up with a multi-disciplinary care team consisting of a cardiologist, heart failure nurse, physiotherapist, and a psychologist at 14 days, 1,2,5 and 12 months post discharge. Telephone counselling by the heart failure nurse and the cardiologist was also available, and for those unable to visit the clinic, home visits were scheduled. Patients had a one-on-one educational session once a month at the patient's home, at the clinic, or by phone with a heart failure nurse. | Patients hospitalized for COPD, with an established diagnosis. | 68.4 | 40.6 |

(Continued)

**Table 1.** (Continued)

| First author, YR | Country | N | Period of Intervention (months) | Follow-Up Period (months) | Description of Usual Care | Description of Intervention | Patient Population | Mean Age of Total Study Population | % Female in Total Study Population |
|---|---|---|---|---|---|---|---|---|---|
| Negarandeh, 2019 [46] | Iran | 80 | 24 | 24 | Patients were discharged given usual care; education was provided by the nurse at this time. | Within the first month, patients were followed-up with twice a week with 20-minute phone calls. Call frequency reduced to one a week during the second month. Phone calls varied by patients' needs and educational questions. The calls evaluated patient's self-care status, providing recommendations and education for performing self-care. | Hospitalized heart failure patients aged 45 or older. | Unable to extract | 33.8 |
| Oscalices, 2019 [47] | Brazil | 201 | 36 | 36 | Patients were followed-up with by the researcher after 3 months. | Patients were given discharge guidance by the researcher. After 30 days, patients were contacted via phone to identify treatment difficulties and clarify doubts. Evaluation of outcomes were performed at 90 days post discharge via telephone. | Patients over the aged of 18 who were admitted to the emergency room with a diagnosis od decompensated heart failure. | 62.6 | 59.2 |
| **COPD Patient Population** | | | | | | | | | |
| Casas, 2006 [41] | Spain, Belgium | 155 | 12 | 12 | Patients were followed-up by their primary care physician usually every 6 months. | Patient follow-up care included extensive assessment and educational sessions by a respiratory nurse regarding COPD and self-monitoring prior to discharge, individual management plans, and availability of a specialized nurse to patients, as well as their caregivers and primary care providers, via telephone. Additionally, patients received at least 1 home visit from the specialized nurse and their primary care team (physician, nurse, and social worker) or solely their primary care provider. During the first month following discharge, weekly telephone calls were made from the nurse to the patients. | Patients presenting with an acute exacerbation of COPD requiring >48 hours of hospitalization. | 71 | 17 |
| Sorknaes, 2013 [40] | Denmark | 266 | 0.25 | 6 | Patients had the option of attending an outpatient clinic with a nurse 1- and 3-months following discharge to confirm the diagnosis of COPD. Thereafter, a plan was made regarding future follow-up. | In addition to usual care and following discharge, patients received daily monitoring via video from a study nurse. Nurse assessments were made during this time and advice was given as needed. Patients measured their heart rate, saturation, and spirometry using their telemonitoring equipment. | Patients >40 years of age admitted to hospital with an acute exacerbation of COPD. | 72 | 61 |

(Continued)

Table 1. (Continued)

| First author, YR | Country | N | Period of Intervention (months) | Follow-Up Period (months) | Description of Usual Care | Description of Intervention | Patient Population | Mean Age of Total Study Population | % Female in Total Study Population |
|---|---|---|---|---|---|---|---|---|---|
| Ritchie, 2016 [38] | USA | 132 | 3 | 3 | Patients received usual post-discharge care by their primary care provider. | Prior to discharge and during the follow-up period, patients received support and education from care transition nurses and an "E-Coach" (IVR-supported care system). This device provided automated telephone calls and was used to accumulate date on patients, provide patients with individualized management plans, and alert the nurse when intervention was required. Patients received calls from the E-Coach everyday for a week following discharge and either everyday or every 3 days for an additional 21 calls. | Patients admitted to hospital with COPD and being discharged home. | 63.6 | 44.9 |
| Ko, 2017 [39] | Hong Kong | 180 | 12 | 12 | Following discharge, patients were advised to continue follow-up with their regular primary care physician. | Prior to discharge, patients participated in educational sessions given by respiratory nurses and physiotherapists. Following discharge, patients were provided with a telephone number that they could call during regular business hours to address questions and/or concerns. Additionally, they received 3 telephone calls per month from a respiratory nurse and were followed-up in clinic by a respiratory specialist every 3 months. | Patients ≥40 years with a diagnosis of COPD admitted with 2 or more of the following: increased dyspnea, increased sputum purulence, increased sputum volume, or one of: nasal discharge/congestion, wheeze, sore throat, cough for at least two days straight. | 74.7 | 4.4 |
| *CKD Patient Population* | | | | | | | | | |
| Ho, 2016 [45] | Taiwan | 106 | 2 | 6 | Regular follow-up with their primary care physician. | Patients were provided with telemonitoring equipment including a pulse oximeter, thermometer, and sphygmomanometer. Patients reported their symptoms daily on an electronic diary for two months post discharge. | No | 80.2 | 23.6 |
| Li, 2014 [42] | China | 135 | 1.5 | 3 | Usual discharge care involved providing patients with information and a telephone hotline service. Patients were then followed-up by their primary care provider. | Patients underwent extensive pre-discharge assessments and education. Following discharge, nurse case managers monitored patients via regular telephone calls starting within 3 days of discharge and occurring once per week for 6 weeks. | Patients hospitalized due to end-stage renal failure. | 56.3 | 41.5 |
| *Undifferentiated High-Risk Patients* | | | | | | | | | |
| Dhalla, 2014 [43] | Canada | 1923 | 12 | 12 | Patients received discharge counseling, a discharge summary (which was also sent to their primary care provider), and prescriptions and home care arrangements as needed. Upon discharge, patients were recommended or scheduled for follow-up appointments with their primary care and specialist physicians. | In addition to usual care, patients received multidisciplinary care in the form of a virtual ward. The virtual ward team consisted of care coordinators, a pharmacist, a nurse or nurse practitioner, a physician, and a clerical assistant. Patients were also monitored via telephone calls, home visits, and/or clinic visits as needed. The VW team held daily meetings to discuss patient care and to design/modify individual treatment plans. | Patients ≥18 years of age being discharged from a general internal medicine ward and at high risk of being readmitted determined by a LACE (length of stay, acuity of the admission, comorbidities, and emergency department visits in the past 6 months) score of ≥10. | 71.3 | 48.5 |

**Table 2. Post-discharge interventions in included studies.**

| First author, YR | Type of Post-Discharge Intervention(s) | | | | |
|---|---|---|---|---|---|
| | Additional Education | Telephone Calls | Home Visit(s) | Tele-monitoring | Multidisciplinary Care |
| *Heart Failure Patient Population* | | | | | |
| Stewart, 1999 [37] | No | No | Yes | No | No |
| Stewart, 1999 [36] | Yes | Yes | Yes | No | Yes |
| Kasper, 2002 [35] | No | Yes | No | No | Yes |
| Goldberg, 2003 [34] | Yes | No | No | Yes | No |
| Dunagan, 2005 [33] | Yes | Yes | No | No | No |
| Cleland, 2005 [32] | No | No | No | Yes | No |
| Cleland, 2005 [32] | No | Yes | No | No | No |
| Del Sindaco, 2007 [31] | Yes | Yes | No | No | Yes |
| Wakefield, 2008 [29] | Yes | No | No | Yes | No |
| Wakefield, 2008 [29] | Yes | Yes | No | No | No |
| Antonicelli, 2008 [30] | Yes | Yes | No | Yes | Yes |
| Giordano, 2009 [28] | Yes | Yes | No | Yes | No |
| Scherr, 2009 [26] | No | No | No | Yes | No |
| Dar, 2009 [27] | Yes | No | Yes | Yes | Yes |
| Leventhal, 2011 [25] | Yes | Yes | Yes | No | No |
| Dendale, 2012 [23] | Yes | No | No | Yes | Yes |
| Angermann, 2012 [24] | Yes | Yes | No | No | No |
| Tsuchihashi-Makaya, 2013 [22] | Yes | Yes | Yes | No | Yes |
| Villani, 2014 [21] | Yes | No | No | Yes | Yes |
| Yu, 2015 [20] | Yes | Yes | Yes | No | No |
| Ritchie, 2016 [38] | Yes | Yes | No | No | No |
| Ong, 2016 [19] | Yes | Yes | No | Yes | No |
| Kotooka, 2018 [17] | No | No | No | Yes | No |
| Negarandeh, 2019 [46] | No | No | No | Yes | No |
| Oscalices, 2019 [47] | Yes | Yes | No | No | No |
| Olivari, 2018 [18] | No | No | No | Yes | No |
| Wierzchowiecki, 2006 [44] | Yes | Yes | Yes | No | Yes |
| *COPD Patient Population* | | | | | |
| Casas, 2006 [42] | Yes | Yes | Yes | No | Yes |
| Sorknaes, 2013 [40] | No | No | No | Yes | No |
| Ritchie, 2016 [38] | Yes | Yes | No | No | No |
| Ko, 2017 [39] | Yes | Yes | No | No | Yes |
| Ho, 2016 [45] | No | No | No | Yes | No |
| *CKD Patient Population* | | | | | |
| Li, 2014 [42] | Yes | Yes | No | No | No |
| *Undifferentiated High-Risk Patients* | | | | | |
| Dhalla, 2014 [43] | No | Yes | Yes | Yes | Yes |

## Studies of heart failure patients

Twenty-five studies [17–38, 44, 46, 47] on heart failure were included in the meta-analysis. Post-discharge interventions reduced the risk of all-cause mortality (RR = 0.73, 95% CI = 0.65–0.83; $I^2$ = 0%; Fig 2A), all-cause hospitalization (RR = 0.90, 95% CI = 0.81–0.99; $I^2$ = 61%; Fig 2B), cause-specific mortality (RR = 0.68, 95% CI = 0.54–0.84; $I^2$ = 0%; Fig 2C), and cause-specific hospitalization (RR = 0.71, 95% CI = 0.63–0.80; $I^2$ = 4%; Fig 2D), in heart failure patients.

## A) All-cause mortality in heart failure studies

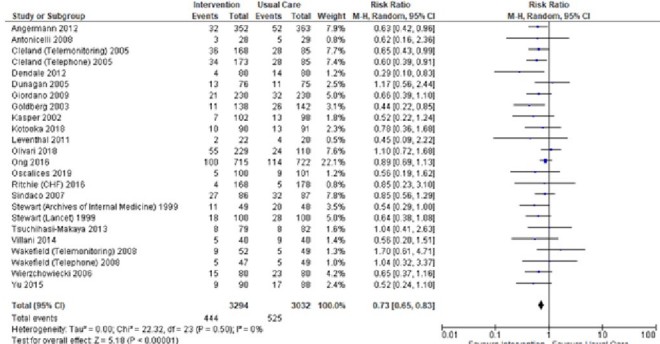

## B) All-cause hospitalization in heart failure studies

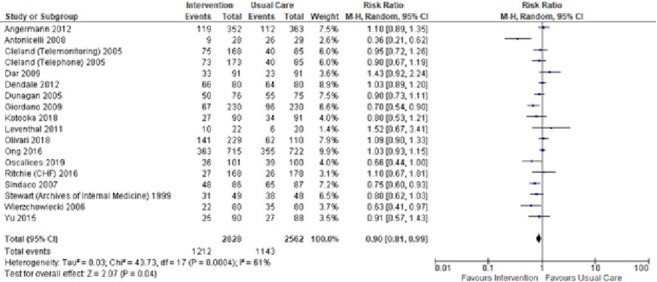

## C) Cause-specific mortality in heart failure studies

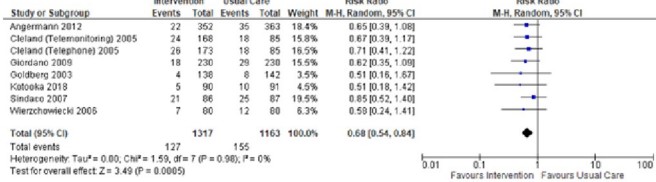

## D) Cause-specific hospitalization in heart failure studies

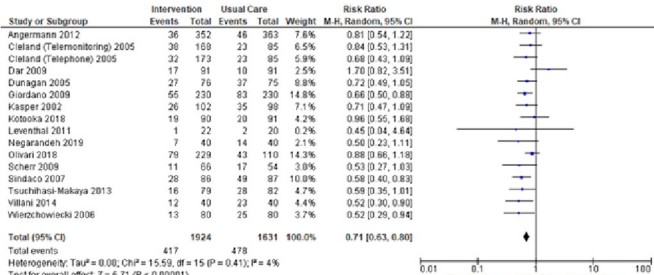

## E) All-cause mortality in COPD studies

## F) All-cause hospitalization in COPD studies

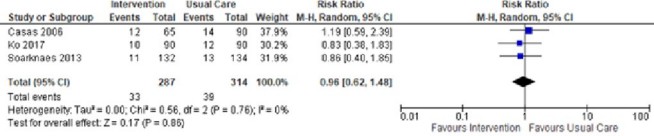

**Fig 2. Meta-analysis of relative risks.** A) all-cause mortality, B) all-cause hospitalization, C) cause-specific mortality, D) cause-specific hospitalization in heart failure patients; E) all-cause mortality, F) all-cause hospitalization in COPD patients.

## Studies of COPD patients

All 5 of the studies examining chronic obstructive pulmonary disease were included in a meta-analysis [38–41, 45]; however, only two outcomes, all-cause mortality and all-cause hospitalization could be analyzed based on the outcomes reported in the studies. Post-discharge interventions reduced the risk of all-cause hospitalization in COPD patients (RR 0.52, 95% CI = 0.32–0.83; $I^2$ = 61%; Fig 2F) yet did not appear to reduce all-cause mortality (RR 0.96, 95% CI = 0.62–1.48; $I^2$ = 0%; Fig 2E).

## Subgroup analyses and meta-regression

In the univariate subgroup analyses, we were unable to discern a statistically significant association between number or type of intervention strategy (multidisciplinary care, pre-discharge patient education, home visits, post-discharge telephone calls and continuous tele-monitoring) and efficacy for any of the outcomes (all-cause and cause-specific hospitalization and all-cause and cause-specific mortality) (Table 3). We did find an association between the reported presence of a trained cardiac nurse and a lower risk for all-cause mortality in heart failure studies [(HR 0.64, 95% CI = 0.54–0.75; $I^2$ = 0%; Fig 3A)] vs. (HR 0.87, 95% CI = 0.73–1.03; $I^2$ = 0%; Fig 3B)]. In a secondary analysis, we used a meta-regression strategy together with a simple scoring system to rank intervention strategies in terms of complexity (pre-discharge patient education = 1, post-discharge telephone calls = 2, home visits = 3, continuous tele-monitoring = 4, and multidisciplinary team care = 5). We defined a complexity score for a study as the simple sum of the complexity ranks of all strategies employed in that study. We were unable to show a clear relationship between complexity score and efficacy for any outcome in the heart-failure studies using meta-regression (Table 4). Additionally, we conducted subgroup analysis including only recently published heart-failure studies (since January 2009). Subgroup and main analysis results were found to be consistent as calculated risk ratios were of similar direction and magnitude in all cases. (S4–S7 Figs).

## Discussion

In this systematic review and meta-analysis of RCT's examining post-discharge interventions in heart failure, COPD and CKD, we found that the interventions studied consistently reduced all-cause mortality, cause-specific mortality, all-cause hospitalization, and cause-specific hospitalization in heart failure patient populations. We were unable to discern an association between complexity (number or type) of intervention strategy employed using the data available. Calculated risk ratios were similar to main analysis when removing older studies in CHF (published prior to 2009), showing no clear difference in effectiveness over time. In a post hoc analysis, inclusion of cardiac specific nurses in the intervention team was associated with greater efficacy in heart failure trials.

Previous systematic reviews have found that interventions including patient education, home visits, self-management support, and telemonitoring have been effective in reducing hospital readmissions [48–52]. As such, these findings support those reported in our previous work [8] and are broadly consistent with recent systematic reviews in heart failure [49–52]. Thus, there appears to be strong evidence for efficacy of post-discharge interventions in heart

**Table 3. Component analysis in heart failure studies.**

| Component Analysis in Congestive Heart Failure Studies | |
|---|---|
| **Multidisciplinary Care** | |
| **Outcome** | **Interaction p-value** |
| All-Cause Mortality | 0.4864 |
| All-Cause Hospitalization | 0.2847 |
| Cause-Specific Mortality | 0.3846 |
| Cause-Specific Hospitalization | 0.2264 |
| **Patient Education** | |
| **Outcome** | **Interaction p-value** |
| All-Cause Mortality | 0.8803 |
| All-Cause Hospitalization | 0.7366 |
| Cause-Specific Mortality | 0.8798 |
| Cause-Specific Hospitalization | 0.2228 |
| **Home Visits** | |
| **Outcome** | **Interaction p-value** |
| All-Cause Mortality | 0.3068 |
| All-Cause Hospitalization | 0.2996 |
| Cause-Specific Mortality | No studies |
| Cause-Specific Hospitalization | 0.4173 |
| **Telephone Calls** | |
| **Outcome** | **Interaction p-value** |
| All-Cause Mortality | 0.7564 |
| All-Cause Hospitalization | 0.1723 |
| Cause-Specific Mortality | 0.6193 |
| Cause-Specific Hospitalization | 0.1257 |
| **Telemonitoring** | |
| **Outcome** | **Interaction p-value** |
| All-Cause Mortality | 0.3658 |
| All-Cause Hospitalization | 0.6938* |
| Cause-Specific Mortality | 0.4975 |
| Cause-Specific Hospitalization | 0.2463 |

* Risk ratio model did not converge, a binomial distribution with a LOGIT link was used to converge model.

failure patients. This research adds to previous systematic reviews by including more recently published studies as well as by analyzing the presence of an era effect in heart failure studies.

The data in COPD and CKD are sparse, and at present only support the efficacy of post-discharge interventions related to all-cause hospitalization in COPD patients. More research is required in these settings.

Other systematic reviews have found that interventions with increased complexity were more effective in reducing hospital readmissions compared to less complex interventions [48]. We could not detect a clear relationship between intervention complexity and efficacy across heart failure studies, nor were we able to identify components of the interventions most responsible for variations in efficacy across studies. This was true both in our primary analysis comparing sub-groups, or in a more sophisticated supplemental analysis using meta-regression and a simple complexity score. One possible interpretation is that there is no true association, and that the effect of a post-discharge intervention on outcomes is driven primarily by increased surveillance and is less dependent on the precise nature of that surveillance. We

### A) All-cause mortality incorporating a cardiac nurse

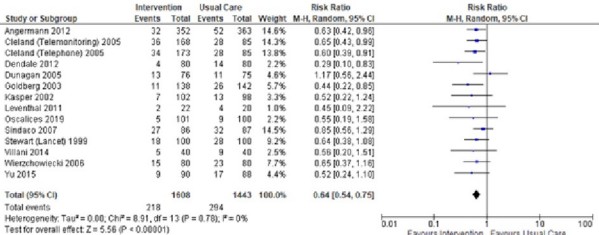

### B) All-cause mortality without a cardiac nurse

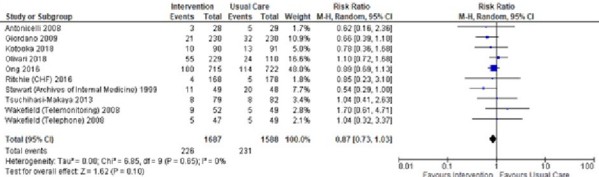

### C) Heart failure mortality incorporating cardiac nurse

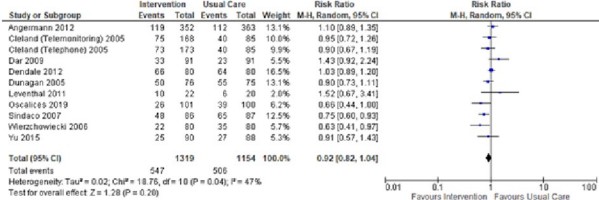

### D) All-cause hospitalization incorporating a cardiac nurse

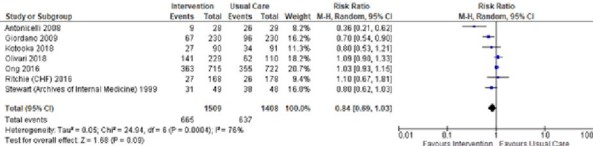

### E) All-cause hospitalization without a cardiac nurse

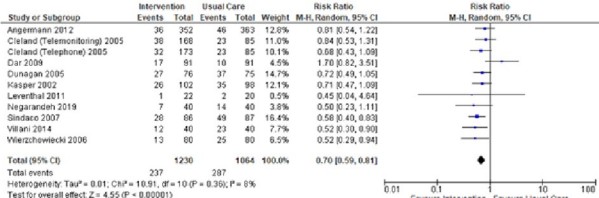

### F) Heart failure hospitalizations incorporating a cardiac nurse

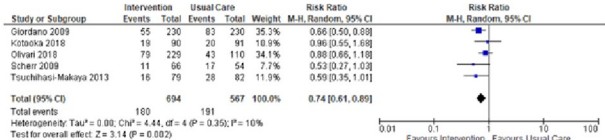

### G) Heart failure hospitalizations without a cardiac nurse

**Fig 3. Subgroup analysis–presence and absence of a cardiac nurse in heart failure studies.**

Table 4. Complexity score analysis in heart failure studies.

| Outcome | P-value |
|---|---|
| All-Cause Mortality | 0.9317* |
| All-Cause Hospitalization | 0.4962 |
| Cause-Specific Mortality | 0.721 |
| Cause-Specific Hospitalization | 0.2423 |

* Risk ratio model did not converge, a binomial distribution with a LOGIT link was used to converge model.

cannot exclude, however, the possibility that such an association exists, but was obscured by the limitations of our approach. In this regard it is important to note that no standard, validated measure of intervention complexity exists. Even our meta-regression strategy using a simple weighting system based on the reported components of an intervention, while reasonable a priori, may have been too crude to capture an association that may well be complex and nonlinear. Given sufficient data points (i.e., individual studies), it might be possible to accurately model the association between intervention complexity and efficacy; however, such techniques are not possible in the context of a meta-analysis of only 23 data points (studies). Indeed, it is unlikely that sufficient data points (i.e., studies) would ever be attainable in the setting of a systematic review and meta-analysis on this topic.

We did observe that studies including a trained, cardiac-specific nurse observed greater risk reduction for mortality and hospitalizations than those without a cardiac specific nurse. Although this finding in isolation must be interpreted cautiously, it is congruent with previous observations that the disease specificity of an intervention is important [8]. Taken together, these findings suggest that a specific-disease focus and expertise is needed in designing and implementing successful post-discharge interventions.

Our study has significant strengths. We adhered to PRISMA recommendations in the conduct of the review [13]. Our search strategy was broad, capturing many randomized clinical trials on diverse types of post-discharge interventions in recently discharged patients with congestive heart failure, chronic kidney disease, or chronic obstructive pulmonary disease. Our analysis included a wide spectrum of interventions that have been implemented in these patient populations in the past two decades. Our review included only randomized clinical trials reporting hard outcomes, because such studies provide the highest level of evidence for the efficacy of interventions.

Our study also has some important limitations. Few studies meeting our inclusion criteria were found for COPD and CKD, limiting conclusions about post-discharge interventions in these conditions. We were unable to show any association between complexity of the intervention and outcome. As discussed above, it may be that our study lacked sufficient power to detect a relationship between these variables if one existed, but it is also probable that this question cannot be answerable by a systematic review of any reasonable size.

Despite these limitations, we believe our findings provide a valid summary of the evidence to date, and as such have implications for research and clinical care. Our results confirm a compelling and consistent benefit for a wide array of post-discharge interventions in heart failure patients. Even though our study was unable to identify a minimum efficacious set of interventions, defining such a subset remains important, as complexity and cost are major barriers to the real-world implementation and scaling of these strategies. Our results suggest that meta-analytic techniques may not be able to answer this question and that randomized clinical trials comparing the efficacy and cost effectiveness of different types of post-discharge intervention should be conducted. In the absence of such data, programs planning on implementing post-

discharge interventions for heart failure are justified in choosing specific interventions based on which of the published strategies appear most feasible in the local context. Our data further suggest that inclusion of a cardiac nurse in that strategy may be critical. Finally, evidence on post-discharge interventions in COPD and CKD are lacking, and further RCT data is urgently needed, particularly for COPD, which is one of the leading chronic diseases requiring readmission [2].

## Conclusions

In conclusion, post-discharge interventions appeared effective in preventing readmission in heart failure populations. Inclusion of a trained cardiac nurse may be an important feature. Additional research is urgently needed on the impact of post-discharge intervention strategies in CKD and COPD.

## Supporting information

**S1 Checklist. PRISMA 2009 checklist.**
(DOC)

**S1 Fig. Meta-analysis of relative risks of all-cause mortality in heart failure patients, excluding those studies deemed at high risk of bias.**
(DOCX)

**S2 Fig. Meta-analysis of relative risks of all-cause hospitalization in heart failure patients, excluding those studies deemed at high risk of bias.**
(DOCX)

**S3 Fig. Meta-analysis of relative risks of cause-specific hospitalization in heart failure patients, excluding those studies deemed at high risk of bias.**
(DOCX)

**S4 Fig. Meta-analysis of relative risks of all-cause hospitalization in heart failure patients, excluding studies published prior to 2009.**
(DOCX)

**S5 Fig. Meta-analysis of relative risks of cause-specific hospitalization in heart failure patients, excluding studies published prior to 2009.**
(DOCX)

**S6 Fig. Meta-analysis of relative risks of all-cause mortality in heart failure patients, excluding studies published prior to 2009.**
(DOCX)

**S7 Fig. Meta-analysis of relative risks of cause-specific mortality in heart failure patients, excluding studies published prior to 2009.**
(DOCX)

**S1 Table. Search strategies.**
(XLSX)

**S2 Table. The Cochrane Collaboration's Tool for randomized studies risk of bias criteria.**
(XLSX)

**S3 Table. Virtual full text review.**
(XLSX)

**S4 Table. Individual risk of bias assessment for all included studies using the Cochrane Collaboration's Tool for randomized studies.**
(XLSX)

## Author Contributions

**Conceptualization:** Ruchi Chhibba, Navdeep Tangri, Paul Komenda, Claudio Rigatto.

**Data curation:** Jenna Sabourin, Domenic Pieroni.

**Formal analysis:** Ryan J. Bamforth, Thomas W. Ferguson.

**Investigation:** Ryan J. Bamforth, Ruchi Chhibba, Claudio Rigatto.

**Methodology:** Ryan J. Bamforth, Thomas W. Ferguson, Nicole Askin, Navdeep Tangri, Paul Komenda, Claudio Rigatto.

**Resources:** Nicole Askin.

**Supervision:** Navdeep Tangri, Paul Komenda, Claudio Rigatto.

**Visualization:** Ryan J. Bamforth.

**Writing – original draft:** Ruchi Chhibba, Thomas W. Ferguson, Jenna Sabourin, Domenic Pieroni, Paul Komenda.

**Writing – review & editing:** Ryan J. Bamforth, Thomas W. Ferguson, Claudio Rigatto.

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
