## [Decision Letter · Decision Letter 0]

14 Sep 2020

PONE-D-20-21147

Strategies to prevent hospital readmission and death in patients with advanced chronic diseases: a systematic review and meta-analysis

PLOS ONE

Dear Dr. Rigatto,

Thank you for submitting your manuscript to PLOS ONE. After careful consideration, we feel that it has merit but does not fully meet PLOS ONE’s publication criteria as it currently stands. Therefore, we invite you to submit a revised version of the manuscript that addresses the points raised during the review process.

Please note that both reviewers expressed important issues to improve the manuscript. I agree with them and would like to to address them and include the suggestions in the revised manuscript.

We look forward to receiving your revised manuscript.

Kind regards,

Hans-Peter Brunner-La Rocca, M.D.

Academic Editor

PLOS ONE

Journal Requirements:

2. In your Methods section, please ensure you have stated the date of last search for studies in this review, ensuring it is up to date enough to allow the inclusion of studies published in the past 12 months.

"We gratefully acknowledge the support of the University of Manitoba Max Rady College of Medicine and the Chronic Disease Innovation Centre, Seven Oaks General Hospital."

Reviewers' comments:

Reviewer's Responses to Questions

**Comments to the Author**

1. Is the manuscript technically sound, and do the data support the conclusions?

Reviewer #1: Yes

Reviewer #2: Yes

2. Has the statistical analysis been performed appropriately and rigorously? 

Reviewer #1: Yes

Reviewer #2: Yes

3. Have the authors made all data underlying the findings in their manuscript fully available?

Reviewer #1: Yes

Reviewer #2: Yes

4. Is the manuscript presented in an intelligible fashion and written in standard English?

Reviewer #1: Yes

Reviewer #2: Yes

5. Review Comments to the Author

Reviewer #1: Strategies to prevent hospital readmission and death in patients with advanced chronic diseases: a systematic review and meta-analysis

Reviewer Recommendation and Comments for Manuscript Number PONE-D-20-21147 from Per Arne Holman.

Summary of the research and your overall impression

I congratulate the authors with a comprehensive literature review on an area of importance. This study aim to identify effective strategies to prevent hospital readmission and deaths in patients with chronic diseases. The focus of review is on three conditions and five strategies. This focus could be mentioned more explicit in the title and motivated more explicit in the introduction. This would help the reader to understand what to expect from the study. The strength of the study is the search method applied on a large number of articles published over the last 20 years. One limitation, not discussed in the paper, is that each strategy may have changed methodologically over so many years.

Discussion of specific areas for improvement

I have one major and one minor issue to address, which in my opinion would improve the manuscript without influencing the results.

Major issue: The article should explain the reasoning more explicit for the scope chosen; three of many possible chronical conditions and the five strategies. In my opinion, it is unclear whether the results depend on the search method focusing particularly on the named five strategies in the chronic condition groups, or if the review identified only the five named strategies after a broad search, or if the five were the most promising of many.

I would recommend to outline whether the literature search starts with, not land on, three patient groups (CHF, COPD, CKD) and five strategies (education, phone, visits, tele monitor and multidisciplinary team) in title or/and very early in the introduction. In the introduction (87) there is an anchor to article reference 8. If the authors give this article some more mentioning, it could help the reader to understand why that article frame the knowledge gap the intend to fill.

From line 99 it would help to understand why you included the particular interventions in the search, and why possible others were left out, cf. line 142 where it is not clear either.

(89) Is virtual wards a focus area in this article already before the systematic review of literature started? If so, should it be included in the title? A closer description of the strategies in the introduction or title would help the reader to understand the scope better; a comparison of “virtual wards” and “less complex and less costly interventions” . (Ref. 142-145) Virtual wards in not mentioned in the results nor in the discussion. Is virtual wards the same as telemonitoring strategy in the result section?

Minor issue: The 20 years long period of inclusion face the challenge that each strategy identified may have changed over time. This is in particularly relevant for the introduction of modern technology in telemedicine in recent years.

The results reported on literature from studies of CHF is based on 25 articles from 1999 to 2018. 15 studies are more than 10 years old. In the discussion I would have expected this article to comment on the fact that telemedicine as an intervention has undergone rapid changes, as indicated in the introduction when speaking of virtual wards; sensor technology (e.g. heart rate, blood pressure, dO2 and weight), PROMIS, video consultations and cellphones. One could expect that such interventions had been more promising and proven more effect on the outcomes in recent years. Apparently, there are few RCT studies on new telemedicine technology available yet. In my opinion this is something to be discussed. At least as an area for future studies. The authors could consider running the test again based on the 10 articles from the last decade, to see if telemedicine in fact produce higher interaction with the outcomes for CHF. I would expect to see changes in the way patient education is delivered to change in a more digital direction.

Miscellaneous remarks

(Abstract, 30, 74) The use of «Rehospitalization». The article use «readmission» mostly, but in the background section rehospitalization is introduced. I suggest a consistent use of readmission will clarify if there is a difference between the two. A definition of a readmission in the introduction would help the reader and might be useful to understand if different definitions may have affected the search of literature performed.

(81) Discuss if 60% of hospital readmissions are preventable.

I’m not an expert, by fare, in systematic literature of meta-analysis. However, I’ve gone through the supplements. The methods are, in my opinion, well described. I recommend the editor to invite peer reviewers with more expertise in this area.

Reviewer #2: This is a well written article on an important topic and clearly a major undertaking. I am concerned that there is no discussion of other systematic reviews on this topic and how this study compares with these. A simple literature research brought up article such as:

Leppin AL, Gionfriddo MR, Kessler M, Brito JP, Mair FS, Gallacher K, et al. Preventing 30-day hospital readmissions: a systematic review and meta-analysis of randomized trials. JAMA Intern Med. 2014;174(7):1095-107.

Gwadry-Sridhar FH, Flintoft V, Lee DS, Lee H, Guyatt GH. A systematic review and meta-analysis of studies comparing readmission rates and mortality rates in patients with heart failure. Arch Intern Med. 2004;164(21):2315-20.

Rice H, Say R, Betihavas V. The effect of nurse-led education on hospitalisation, readmission, quality of life and cost in adults with heart failure. A systematic review. Patient Educ Couns. 2018;101(3):363-74.

This study would benefit from a discussion of how this paper adds to previous systematic reviews such as those listed above.

Please briefly discuss why CHF, COPD and CKD were chosen - I understand that these are the commonest conditions with heavy burden but it would be good to have some background to that for the reader.

I could not see a time line for the search in the paper - on reviewing the search criteria - it appeared to be 2015-current yet there are studies from 1999 included - please clarify and state the timeline in the paper.

Table 1 first paper intervention period -0.25 - this must mean one week but is a little confusing perhaps put 0.25(1w) for clarification.

Line 268 - there are two "ins"

6. PLOS authors have the option to publish the peer review history of their article (what does this mean?). If published, this will include your full peer review and any attached files.

Reviewer #1: **Yes: **Per Arne Holman

Reviewer #2: No

---

## [Author Response · Author response to Decision Letter 0]

2 Feb 2021

Dear Dr. Heber

Thank you for considering our manuscript for publication in PLOS One. Please find below our response to editors’ and reviewers’ comments 

Journal Requirements:

Response: Thanks you for the recommendation. Style requirements have been altered to meet journal requirements.

2. In your Methods section, please ensure you have stated the date of last search for studies in this review, ensuring it is up to date enough to allow the inclusion of studies published in the past 12 months.

Response: Thank you, this has been included on page 5, line 110 and reads “We searched MEDLINE, PubMed, Cochrane, EMBASE and CINAHL identifying relevant studies published up to November, 2020”.

"We gratefully acknowledge the support of the University of Manitoba Max Rady College of Medicine and the Chronic Disease Innovation Centre, Seven Oaks General Hospital."

Response: Thank you, we have included a revision in the cover letter and removed the statement from the manuscript.

Reviewers' comments:

Reviewer #1: Strategies to prevent hospital readmission and death in patients with advanced chronic diseases: a systematic review and meta-analysis

Reviewer Recommendation and Comments for Manuscript Number PONE-D-20-21147 from Per Arne Holman.

Summary of the research and your overall impression

I congratulate the authors with a comprehensive literature review on an area of importance. This study aim to identify effective strategies to prevent hospital readmission and deaths in patients with chronic diseases. The focus of review is on three conditions and five strategies. This focus could be mentioned more explicit in the title and motivated more explicit in the introduction. This would help the reader to understand what to expect from the study. The strength of the study is the search method applied on a large number of articles published over the last 20 years. One limitation, not discussed in the paper, is that each strategy may have changed methodologically over so many years.

Response: Thank you for the recommendation, we have altered the title to the following: “Strategies to prevent hospital readmission and death in patients with chronic heart failure, chronic obstructive pulmonary disease and chronic kidney disease: a systematic review and meta-analysis”. 

Discussion of specific areas for improvement

I have one major and one minor issue to address, which in my opinion would improve the manuscript without influencing the results.

Major issue: The article should explain the reasoning more explicit for the scope chosen; three of many possible chronical conditions and the five strategies. In my opinion, it is unclear whether the results depend on the search method focusing particularly on the named five strategies in the chronic condition groups, or if the review identified only the five named strategies after a broad search, or if the five were the most promising of many.

I would recommend to outline whether the literature search starts with, not land on, three patient groups (CHF, COPD, CKD) and five strategies (education, phone, visits, tele monitor and multidisciplinary team) in title or/and very early in the introduction. In the introduction (87) there is an anchor to article reference 8. If the authors give this article some more mentioning, it could help the reader to understand why that article frame the knowledge gap the intend to fill.

From line 99 it would help to understand why you included the particular interventions in the search, and why possible others were left out, cf. line 142 where it is not clear either.

(89) Is virtual wards a focus area in this article already before the systematic review of literature started? If so, should it be included in the title? A closer description of the strategies in the introduction or title would help the reader to understand the scope better; a comparison of “virtual wards” and “less complex and less costly interventions” . (Ref. 142-145) Virtual wards in not mentioned in the results nor in the discussion. Is virtual wards the same as telemonitoring strategy in the result section?

Response: Thank you for the recommendation. Our literature search began with the three patients groups and after reviewing the literature we identified the 5 types of post-discharge interventions. Additionally, we have altered the title to the following: “Strategies to prevent hospital readmission and death in patients with chronic heart failure, chronic obstructive pulmonary disease and chronic kidney disease: a systematic review and meta-analysis”. 

Minor issue: The 20 years long period of inclusion face the challenge that each strategy identified may have changed over time. This is in particularly relevant for the introduction of modern technology in telemedicine in recent years.

The results reported on literature from studies of CHF is based on 25 articles from 1999 to 2018. 15 studies are more than 10 years old. In the discussion I would have expected this article to comment on the fact that telemedicine as an intervention has undergone rapid changes, as indicated in the introduction when speaking of virtual wards; sensor technology (e.g. heart rate, blood pressure, dO2 and weight), PROMIS, video consultations and cellphones. One could expect that such interventions had been more promising and proven more effect on the outcomes in recent years. 

Response: Thank you for the recommendation. We have included additional analyses which excludes studies in heart failure published prior to 2009. The results remain consistent with calculated risk ratios being of similar direction and magnitude.

Miscellaneous remarks

(Abstract, 30, 74) The use of «Rehospitalization». The article use «readmission» mostly, but in the background section rehospitalization is introduced. I suggest a consistent use of readmission will clarify if there is a difference between the two. A definition of a readmission in the introduction would help the reader and might be useful to understand if different definitions may have affected the search of literature performed.

Response: Thanks you for the recommendation. Line 19 has been changed to read readmission rather than rehospitalization.

(81) Discuss if 60% of hospital readmissions are preventable.

Response: Thank you for the recommendation. This has been addressed no page 4, line 74 and reads the following It has been estimated that nearly 60% of hospital readmissions are preventable (4). Risk factors associated with avoidable readmissions include those related to patient, social, clinical and system factors such as patient behaviors, community services, adequacy and appropriateness of assessment and treatment as well as accessibility and coordination within the healthcare delivery system (4).

I’m not an expert, by fare, in systematic literature of meta-analysis. However, I’ve gone through the supplements. The methods are, in my opinion, well described. I recommend the editor to invite peer reviewers with more expertise in this area.

Reviewer #2: This is a well written article on an important topic and clearly a major undertaking. I am concerned that there is no discussion of other systematic reviews on this topic and how this study compares with these. A simple literature research brought up article such as:

Leppin AL, Gionfriddo MR, Kessler M, Brito JP, Mair FS, Gallacher K, et al. Preventing 30-day hospital readmissions: a systematic review and meta-analysis of randomized trials. JAMA Intern Med. 2014;174(7):1095-107.

Gwadry-Sridhar FH, Flintoft V, Lee DS, Lee H, Guyatt GH. A systematic review and meta-analysis of studies comparing readmission rates and mortality rates in patients with heart failure. Arch Intern Med. 2004;164(21):2315-20.

Rice H, Say R, Betihavas V. The effect of nurse-led education on hospitalisation, readmission, quality of life and cost in adults with heart failure. A systematic review. Patient Educ Couns. 2018;101(3):363-74.

This study would benefit from a discussion of how this paper adds to previous systematic reviews such as those listed above.

Response: Thank you for the recommendation. This has been addressed on page 46 line 336 and reads Previous systematic reviews have found that interventions including patient education, home visits, self-management support, and telemonitoring have been effective in reducing hospital readmissions (48-52). As such, these findings support those reported in our previous work (8) and are broadly consistent with recent systematic reviews in heart failure (49-52). Thus, there appears to be strong evidence for efficacy of post-discharge interventions in heart failure patients. This research adds to previous systematic reviews by including more recently published studies as well as by analyzing the presence of an era effect in heart failure studies”. Additionally, we have altered line 350 on page 46 to read ” Other systematic reviews have found that interventions with increased complexity were more effective in reducing hospital readmissions compared to less complex interventions:.

Please briefly discuss why CHF, COPD and CKD were chosen - I understand that these are the commonest conditions with heavy burden but it would be good to have some background to that for the reader.

Response: Thank you for the comment. We have address this on Page 6, line 120 and it reads “We focused specifically on CKD, HF, and COPD as these chronic diseases place heavy burdens on modern health systems with respect to prevalence, resource utilization and costs”.

I could not see a time line for the search in the paper - on reviewing the search criteria - it appeared to be 2015-current yet there are studies from 1999 included - please clarify and state the timeline in the paper.

Response: Thank you for bringing this to our attention. We have amended the manuscript on page 5, line 110 which states the timeline. The initial search went back as far as each database goes with it ending in November, 2020.

Table 1 first paper intervention period -0.25 - this must mean one week but is a little confusing perhaps put 0.25(1w) for clarification.

Response: Thank you for the recommendation, The intervention period in table 1 has been amended to read 0.25 (1 week).

Line 268 - there are two "ins"

Response: Thank you for the comment. The duplicate “in” has been removed from the text at line 268.

Sincerely,

Dr. Claudio Rigatto

Head, Section of Nephrology, University of Manitoba

Associate Medical Director, Manitoba Renal Program 

Seven Oaks General Hospital 

crigatto@sbgh.mb.ca

204-632-3383

On behalf of: 

Ryan J. Bamforth, Ruchi, Chhibba, Thomas W. Ferguson, Jenna Sabourin, Domenic Pieroni, Nicole Askin, Navdeep Tangri and Paul Komenda.

---

## [Decision Letter · Decision Letter 1]

25 Feb 2021

PONE-D-20-21147R1

Strategies to prevent hospital readmission and death in patients with chronic heart failure, chronic obstructive pulmonary disease and chronic kidney disease: a systematic review and meta-analysis

PLOS ONE

Dear Dr. Rigatto,

Thank you for submitting your manuscript to PLOS ONE. After careful consideration, we feel that it has merit but does not fully meet PLOS ONE’s publication criteria as it currently stands. Therefore, we invite you to submit a revised version of the manuscript that addresses the points raised during the review process.

There are some remaining issues that need to be addressed before your paper can be accepted for publication. Please consider to have your manuscript copy-edited by a native speaking person as there is not such editing for PLOS ONE.

We look forward to receiving your revised manuscript.

Kind regards,

Hans-Peter Brunner-La Rocca, M.D.

Academic Editor

PLOS ONE

Journal Requirements:

Reviewers' comments:

Reviewer's Responses to Questions

**Comments to the Author**

1. If the authors have adequately addressed your comments raised in a previous round of review and you feel that this manuscript is now acceptable for publication, you may indicate that here to bypass the “Comments to the Author” section, enter your conflict of interest statement in the “Confidential to Editor” section, and submit your "Accept" recommendation.

Reviewer #1: All comments have been addressed

Reviewer #2: (No Response)

2. Is the manuscript technically sound, and do the data support the conclusions?

Reviewer #1: Yes

Reviewer #2: Yes

3. Has the statistical analysis been performed appropriately and rigorously? 

Reviewer #1: Yes

Reviewer #2: Yes

4. Have the authors made all data underlying the findings in their manuscript fully available?

Reviewer #1: Yes

Reviewer #2: Yes

5. Is the manuscript presented in an intelligible fashion and written in standard English?

Reviewer #1: Yes

Reviewer #2: No

6. Review Comments to the Author

Reviewer #1: (No Response)

Reviewer #2: Thank your for addressing the issues raised in the first review. There are still a few minor typos and punctuation issues so please review for this.

My only concern is that you have addressed why you chose the clinical areas for review CHF, COPD and CKD but this is in the methods - I think it need to be in the introduction immediately after you state the clinical areas reviewed, especially as you have added references which are not really appropriate in the methods section, but would work well in the introduction.

That is - this sentence needs to be in the intro.

“We focused specifically on CKD, HF, and COPD as these chronic diseases place heavy burdens on modern health systems with respect to prevalence, resource utilization and costs”.

Other wise I have no other comments and this is Avery useful addition to the literature.

7. PLOS authors have the option to publish the peer review history of their article (what does this mean?). If published, this will include your full peer review and any attached files.

Reviewer #1: **Yes: **Per Arne Holman

Reviewer #2: **Yes: **Professor Carol J. Peden

---

## [Author Response · Author response to Decision Letter 1]

16 Mar 2021

March 16, 2021

Dr. Joerg Heber

Editor-in-Chief, PLOS One

Dear Dr. Heber

Thank you for considering our manuscript for publication in PLOS One. Please find below our response to editors’ and reviewers’ comments 

Journal Requirements:

Response:

Thank you for outlining this. The reference list has been reviewed for retracted papers as well as formatted to be complete and correct.

Reviewers' comments:

Reviewer #2: Thank your for addressing the issues raised in the first review. There are still a few minor typos and punctuation issues so please review for this.

My only concern is that you have addressed why you chose the clinical areas for review CHF, COPD and CKD but this is in the methods - I think it need to be in the introduction immediately after you state the clinical areas reviewed, especially as you have added references which are not really appropriate in the methods section, but would work well in the introduction.

That is - this sentence needs to be in the intro.

“We focused specifically on CKD, HF, and COPD as these chronic diseases place heavy burdens on modern health systems with respect to prevalence, resource utilization and costs”.

Response:

Thank you for the recommendation. We removed the sentence from the methods section and added the following sentence to the introduction on page 5 line 100:

“We chose these conditions because all of them are common and associated with frequent and costly exacerbations/acute decompensations necessitating admission for stabilization and associated with high risk of recurrence.”

Additionally, the manuscript has been edited to address all spelling, punctuation and grammar errors. We thank you for bringing this up.

Sincerely,

Dr. Claudio Rigatto

Head, Section of Nephrology, University of Manitoba

Associate Medical Director, Manitoba Renal Program 

Seven Oaks General Hospital 

crigatto@sbgh.mb.ca

204-632-3383

On behalf of: 

Ryan J. Bamforth, Ruchi, Chhibba, Thomas W. Ferguson, Jenna Sabourin, Domenic Pieroni, Nicole Askin, Navdeep Tangri and Paul Komenda.

---

## [Editor Report · Decision Letter 2]

22 Mar 2021

Strategies to prevent hospital readmission and death in patients with chronic heart failure, chronic obstructive pulmonary disease, and chronic kidney disease: a systematic review and meta-analysis

PONE-D-20-21147R2

Dear Dr. Rigatto,

We’re pleased to inform you that your manuscript has been judged scientifically suitable for publication and will be formally accepted for publication once it meets all outstanding technical requirements.

Kind regards,

Hans-Peter Brunner-La Rocca, M.D.

Academic Editor

PLOS ONE
---

## [Editor Report · Acceptance letter]

12 Apr 2021

PONE-D-20-21147R2 

Strategies to prevent hospital readmission and death in patients with chronic heart failure, chronic obstructive pulmonary disease, and chronic kidney disease: a systematic review and meta-analysis  

Dear Dr. Rigatto:

I'm pleased to inform you that your manuscript has been deemed suitable for publication in PLOS ONE. Congratulations! Your manuscript is now with our production department. 

Kind regards, 

on behalf of

Dr. Hans-Peter Brunner-La Rocca 

Academic Editor

PLOS ONE